# Bayesian polygenic risk estimation approach to nuclear families with discordant sib-pairs for myelomeningocele

**Adolfo Aguayo-Gómez**, **Leonora Luna-Muñoz**, **Yevgeniya Svyryd**,
**Luis Ángel Muñoz-Téllez**, **Osvaldo M. Mutchinick***

Department of Genetics, National Institute of Medical Sciences and Nutrition Salvador Zubirán (INCMNSZ), Mexico City, Mexico

* osvaldo@unam.mx

**Data Availability Statement:** All relevant data are within the manuscript and its Supporting information files.

## Abstract

Myelomeningocele (MMC) is the most severe and disabling form of spina bifida with chronic health multisystem complications and social and economic family and health systems burden. In the present study, we aimed to investigate the genetic risk estimate for MMC in a cohort of 203 Mexican nuclear families with discordant siblings for the defect. Utilizing a custom Illumina array, we analyzed 656 single nucleotide polymorphisms (SNPs) of 395 candidate genes to identify a polygenic risk profile for MMC. Through a family-based analysis employing the transmission disequilibrium test (TDT) and Bayesian analysis, we assessed risk alleles transmission and calculated conditional probabilities estimating a polygenic risk for MMC. Our findings reveal significant associations of six genes related to neural tube closure (*PSMB4*, *ATIC*, *DKK2*, *PSEN2*, *C2CD3*, and *PLCB2*), showing differences in risk allele transmission between affected and unaffected siblings. Bayesian analysis identified changes in the risk profile after initiating folic acid fortification in Mexico, showing an evident decline in the conditional risk from 1/156 to 1/304 respectively. Despite the decline, this represents a 5.84-fold increase in risk before fortification and a 2.99-fold increase post-fortification compared to the baseline risk level (1/910). Our study highlights the advantage of incorporating a Bayesian analytical methodology in families with discordant sib-pairs, offering insights into the polygenic risk estimate for MMC and, most probably, for other congenital malformations.

## Introduction

Myelomeningocele (MMC), the severe form of spina bifida, is the most common and disabling neural tube defects (NTDs) [1]. Worldwide prevalence varies from 0.5 to 10 per 1,000 pregnancies [2], having the Mexican population in the near past one of the world's highest prevalences of NTD, with approximately 1/250 conceptions reaching 20 weeks of gestation [3], remaining high, despite a persisting decrease in the prevalence at birth over the last three decades. According to the 2019 data from our Registry and Epidemiological Surveillance of

**Funding:** OMM CONACYT-SALUD-2013-1-201547 CONSEJO NACIONAL DE CIENCIA Y TECNOLOGÍA CONACYT.GOB.MX THE SPONSOR DID NOT PLAY ANY ROLE IN THE STUDY DESIGN, DATA COLLECTION AND ANALYSIS, DECISION TO PUBLISH, OR PREPARATION OF THE MANUSCRIPT.

**Competing interests:** The authors declare that no competing interests exist.

Congenital Malformations (RYVEMCE) program, the prevalence of MMC is 1/1,052 births [4–7]. The etiology of this birth defect is multifactorial, involving complex interactions of genetic and environmental risk factors [1,8], affecting the intrinsic mechanisms of neural tube closure during early embryonic development [9]. While over 200 NTD-causing genes for MMC and other NTDs were identified in mouse models, only a few candidate gene risk variants have been associated with MMC in humans [10]. Folic acid deficiency is critical in its occurrence among maternal-nutritional risk factors, as evidenced by a 72% reduction in risk recurrence (RR, 0.28; 95% CI, 0.12, 0.71) in subsequent pregnancies [11]. Other environmental factors, such as maternal obesity, diabetes, and exposure to certain medications (e.g., trimethoprim, valproic acid, and methotrexate), have also been associated with an increased risk for the defect [12–14].

Family-based studies of candidate genes have been frequently used to investigate the genetic predisposition concerned with the etiology of spina bifida and other NTDs [15]. One advantage of family-based studies is their ability to control confounders such as population stratification; however, these studies require more resources and larger sample sizes than case-control designs [16]. Among nuclear family studies, the transmission disequilibrium test (TDT) is a methodology based on nuclear family trios involving the affected child and parents to investigate the relationship with gene risk variants [17]. This method and its variants have proven to be valuable in the context of candidate gene studies in complex diseases like MMC [18]. This test is also valuable for *quartets*, adding to the trio a non-affected (discordant) sib comparing the transmission from an informative mating of parents (at least one of them heterozygous) of a particular risk allele to the affected and unaffected sib [19,20]. So, if a risk allele is linked with the disease, it will be significantly more frequently transmitted to affected siblings than to the unaffected ones. This methodology has various advantages by inducing harmonization as discordant sibs share similar genetic backgrounds and environmental exposures, helping control the effect of parental health, maternal nutrition habits, and other environmental factors [21–23]. Family-based studies on genetic variants associated with unspecified spina bifida and MMC usually included limited gene variants, focusing mainly on trios [10]. Recent research has revealed the involvement of genes in Hedgehog (Hh), Wnt, planar cell polarity (PCP), and ciliogenesis signaling pathways, as well as genes related to folate, glucose, fatty acids, and SOD1-SOD2 metabolism [24–26].

Although before 2010, Greene et al. provided a comprehensive anthology of family-based studies in this field [10], to our knowledge, no studies on NTDs, mainly MMC, have employed the discordant sib-pairs approach within a family-based study framework. Despite recent progress focused on rare variants from exome [27–30] or genome studies [31–33] in spina bifida, the causes, mechanisms, and risks are not fully understood. Similarly, family-based candidate gene studies may bring more light to a better understanding the complex genetics causing NTDs. These methods can potentially identify the effect of multiple gene variants associated with MMC, providing valuable insights into their genetic predisposition.

Our study aimed to investigate the polygenic risk profile for MMC in a cohort of 203 informative Mexican nuclear families with discordant sibs for the defect by analyzing 656 gene variants of 395 candidate genes using a Bayesian analysis approach.

## Methodology

### Population studied

A sample of 203 discordant sib-pair-parents informative family quartets was selected from a multicenter population-based study of 1030 families with different family structures (trios, quartets, and duos). Each family comprised a child diagnosed with MMC, his/her unaffected

sib, and their molecular-confirmed biological parents. Participating families came from 11 Teleton's Infant Rehabilitation Centers from 11 different cities in 10 different states of México. The recruitment period was from September 5, 2013, to September 4, 2014. Pediatricians and geneticists confirmed MMC diagnosis through clinical and X-ray studies. Only isolated MMC cases were included in the studied sample. Written informed consent was obtained from the parents, and the Ethics and Research Committee and Research Committee approved the development of the study (GEN-340-13/15-2BIS).

## Genotyping

Genotyping of known single nucleotide polymorphisms (SNPs) was performed in extracted DNA from peripheral blood or oral mucosa samples following QIAamp® DNA mini kit protocols (Hilden, Germany) with a custom Illumina GoldenGate array (Illumina®, San Diego, CA) designed for the analysis of 768 SNPs, 656 from 395 candidate genes for neural tube defects.

## Statistical analysis

**Allele and genotype frequencies.** We carried out the analysis of allelic and genotypic frequencies, including the Hardy-Weinberg equilibrium test, to assess if genotype frequencies deviated significantly ($p < 0.05$) from it. Linkage disequilibrium between pairs of variants was searched using PLINK and missing genotype rates per individual, and variants filtering using specified threshold values to ensure data quality. We excluded variants or samples from subsequent analyses that significantly deviated from Hardy-Weinberg equilibrium ($p < 0.05$), had a call rate below 98%, displayed a genotyping quality score below 90, or exhibited Mendelian errors.

**Family-based analysis.** Using the TDT, we first analyzed the presence or not of overtransmission of parental risk alleles to their MMC offspring group. Similarly, TDT analysis was run for healthy siblings to evaluate the frequency of whether the risk or wildtype allele variants were transmitted to the group of sibs. Additionally, we incorporate TDT analysis of discordant sib-pairs (TDT-DS) into the MMC familial quartets, employing the methodology outlined by Deng et al. [20]. Furthermore, McNemar's test for the discordant sib-pairs was applied to informative families. We also used conditional logistic regression models for discordant sibling pairs (S1 Table). The analyses performed encompassed: 1) the analysis of TDT in trios of cases and siblings detecting 10 of 656 candidate genes allele variants in 10 different genes exhibiting significant differences for a nominal p-value of <0.05; and 2) evaluation of this selected group of variants for conditional probabilities estimations through Bayesian analysis.

**Bayesian analysis.** The Bayesian analysis compared the probability of risk allele variants in MMC cases and their unaffected siblings, incorporating prior known MMC prevalence and probabilistic modeling to estimate the risk associated with specific allele transmissions. The analysis also explored the joint effects of multiple variants, focusing on their combined effect on the congenital defect risk to occur. For this analysis, we considered an *a priori* probability (prevalence at birth) of isolated spina bifida of 1.098/1,000 or 1/910 births, according to RYVEMCE data from 1978 to 2002, and a prevalence of 0.562/1,000 or 1/1,779 births from 2003 to 2019, corresponding to the specific periods before (pre-FAFP) and after (post-FAFP) folic acid fortification in Mexico [34]. Further details on the Bayesian modeling approach, including the full Bayes' Theorem framework used, parameter priors, and the assumptions regarding independent variant effects, are included in (S1 File).

**Variant prioritization.**   The analyses were conducted by categorizing gene variants into missense and non-missense variants (intron and synonymous). Further analysis included all ten variants. This approach facilitated a thorough examination of the potential risk interactions among the different types of gene variants and the outcomes under investigation. Only one variant per gene was selected for analysis when two or more variants were present within a single gene. This approach aimed to prevent potential biases and reduce multicollinearity that could arise from analyzing multiple highly correlated variants within the same gene.

Furthermore, independent effects are assumed when combining the effects of multiple variants. This approach estimates the combined polygenic risk of an embryo developing MMC from a zygote exposed simultaneously to various risk variants resulting from the risk of allele transmissions to affected and non-affected respective siblings. The supplementary methods file shows more detailed descriptions of the variant prioritization procedure and the assumptions of independent variant effects. All the analyses were realized using PLINK (www.cog-genomics.org/plink/1.9/), STATA 12 software packages, and the R program, in which a specific implementation for the analysis of these data on an interactive Shiny application was developed (https://www.rstudio.com/products/shiny/), (S2 and S3 Files).

## Results

### Risk gene variants detection

In the studied sample of 203 families, we identified genetic risk variants (GRV) in 10 genes *MTHFR* (c.677C>T), *PSMB4* (c.701T>C), *PSEN2* (c.261C>T), *ATIC* (c.347C>G), *DKK2* (c.437G>A), *MTRR* (c.1049A>G), *TFAP2A* (c.46-1620A>G), *C2CD3* (c.4923A>G), *PLCB2* (c.1182C>T), and *SLC7A6OS* (c.134G>A). Of these, six were missense variants, three were synonymous, and one was intronic (Table 1).

**Table 1. Characteristics of the 10 included genetic variants analyzed.**

| GENE | CHR | LOCATION (hg19) | RefSNP | CONSEQUENCE | EXON | INTRON | HGVSc | HGVSp | AMINO ACID | CODON CHANGE |
|---|---|---|---|---|---|---|---|---|---|---|
| *MTHFR* | 1 | 11856378 | rs1801133 | MISSENSE | 4/11 | - | ENST00000376592.1: c.677C>T | ENSP00000365777.1:p. Ala222Val | A/V | gCc/gTc |
| *PSMB4* | 1 | 151374025 | rs4603 | MISSENSE | 6/7 | - | ENST00000290541.7: c.701T>C | ENSP00000290541.6:p. Ile234Thr | I/T | aTc/aCc |
| *ATIC* | 2 | 216190020 | rs2372536 | MISSENSE | 5/16 | - | ENST00000236959.9: c.347C>G | ENSP00000236959.9:p. Thr116Ser | T/S | aCt/aGt |
| *DKK2* | 4 | 107845794 | rs17037102 | MISSENSE | 3/4 | - | ENST00000285311.3: c.437G>A | ENSP00000285311.3:p. Arg146Gln | R/Q | cGg/cAg |
| *MTRR* | 5 | 7885959 | rs162036 | MISSENSE | 7/15 | - | ENST00000264668.2: c.1130A>G | ENSP00000264668.2:p. Lys377Arg | K/R | aAg/aGg |
| *SLC7A6OS* | 16 | 68344696 | rs3803650 | MISSENSE | 1/5 | - | ENST00000263997.6: c.134G>A | ENSP00000263997.5:p. Gly45Asp | G/D | gGt/gAt |
| *PSEN2* | 1 | 227071525 | rs1046240 | SYNONYMOUS | 5/13 | - | ENST00000366783.3: c.261C>T | ENSP00000355747.3:p. His87 = | H | caC/caT |
| *C2CD3* | 11 | 73785326 | rs4453265 | SYNONYMOUS | 24/31 | - | ENST00000313663.7: c.4923A>G | ENSP00000323339.7:p. Val1641 = | V | gtA/gtG |
| *PLCB2* | 15 | 40590134 | rs2229690 | SYNONYMOUS | 12/32 | - | ENST00000260402.3: c.1182C>T | ENSP00000260402.3:p. Ser394 = | S | agC/agT |
| *TFAP2A* | 6 | 10412188 | rs1675414 | INTRONIC | - | 1/6 | ENST00000379604.2: c.46-1620A>G | - | - | - |

CHR, Chromosome; HGVSc, HGVS transcript nomenclature; HGVSp, HGVS protein nomenclature.

## TDT analysis

Table 2 exhibits the results of the TDT trio analysis of the ten risk gene variants that showed significant statistical differences (SSD). The *PSMB4* c.701T>C variant showed a significant excess of risk allele transmissions in MMC cases (OR of 1.39 (95% CI 1.01–1.92), p = 0.036, and significantly lower transmission of the risk allele in siblings (OR of 0.67 (95% CI: 0.49–0.91), p = 0.010. For the remaining variants, the ORs in MMC cases ranged from 1.36 observed for the *MTRR* c.1049A>G variant to 1.81 for the *PLCB2* c.1182C>T. Except for the *PSMB4* variant, there were no other significant deviations in the expected allele transmission in the healthy siblings group.

The results of the TDT analysis, including the sib-pair model (Table 3), showed SSD in six gene variants, three missense, and three synonymous. The first three, *PSMB4* (c.701T>C), *ATIC* (c.347C>G), and *DKK2* (c.437G>A) with p-values of 0.001, 0.007, and 0.023, respectively. The three synonymous *PLCB2* (c.1182C>T), *PSEN2* (c.261C>T), and *C2CD3* (c.4923A>G), with p-values of <0.001, 0.034, and 0.030, respectively. Although higher frequencies of risk allele transmission were observed in the *MTHFR* (c.677C>T), *MTRR* (c.1049A>G), *SLC7A6OS* (c.134G>A), and *TFAP2A* (C.46-1620A>G) gene variants, they did not reach SSD.

## Bayesian analysis results

Conditional probabilities were also estimated for the pre-FAFP (1978–2002) and the post-FAFP (2003–2019). These periods were distinctively selected to correlate with the initiation of

**Table 2. Results of Transmission Disequilibrium Test (TDT) analysis in MMC cases and healthy siblings in 203 family quartets.**

| GENE | RefSNP | DNA CHANGE | GROUP | IT | TRANSMITTED ALLELES | | | UNTRANSMITTED ALLELES | | | OR (95% CI) | P-value |
|---|---|---|---|---|---|---|---|---|---|---|---|---|
| | | | | | Allele | O | E | Allele | O | E | | |
| *PSMB4* | rs4603 | c.701T>C | MMC | 165 | C | 96 | 82.5 | T | 69 | 82.5 | 1.39 (1.01–1.92) | 0.0356* |
| | | | SIBLINGS | 165 | C | 66 | 82.5 | T | 99 | 82.5 | 0.67 (0.48–0.92) | 0.0102* |
| *ATIC* | rs2372536 | c.347C>G | MMC | 173 | G | 106 | 86.5 | C | 67 | 86.5 | 1.58 (1.15–2.18) | 0.0030* |
| | | | SIBLINGS | 173 | G | 80 | 86.5 | C | 93 | 86.5 | 0.86 (0.63–1.17) | 0.3230 |
| *DKK2* | rs17037102 | c.437G>A | MMC | 172 | A | 101 | 86 | G | 71 | 86 | 1.42 (1.04–1.96) | 0.0222* |
| | | | SIBLINGS | 172 | A | 79 | 86 | G | 93 | 86 | 0.85 (0.62–1.16) | 0.2858 |
| *MTHFR* | rs1801133 | c.677C>T | MMC | 178 | T | 107 | 89 | C | 71 | 89 | 1.51 (1.11–2.06) | 0.0070* |
| | | | SIBLINGS | 178 | T | 93 | 89 | C | 85 | 89 | 1.09 (0.81–1.49) | 0.5488 |
| *MTRR* | rs162036 | c.1049A>G | MMC | 191 | G | 110 | 95.5 | A | 81 | 95.5 | 1.36 (1.01–1.83) | 0.0359* |
| | | | SIBLINGS | 191 | G | 91 | 95.5 | A | 100 | 95.5 | 0.91 (0.68–1.22) | 0.5149 |
| *SLC7A6OS* | rs3803650 | c.134G>A | MMC | 166 | A | 96 | 83 | G | 70 | 83 | 1.37 (1.01–1.87) | 0.0436 |
| | | | SIBLINGS | 167 | A | 91 | 83.5 | G | 76 | 83.5 | 1.20 (0.88–1.62) | 0.2460 |
| *PSEN2* | rs1046240 | c.261C>T | MMC | 178 | G | 103 | 89 | A | 75 | 89 | 1.37 (1.02–1.85) | 0.0358* |
| | | | SIBLINGS | 178 | G | 82 | 89 | A | 96 | 89 | 0.85 (0.64–1.15) | 0.2940 |
| *C2CD3* | rs4453265 | c.4923A>G | MMC | 189 | G | 110 | 94.5 | A | 79 | 94.5 | 1.39 (1.04–1.86) | 0.0241* |
| | | | SIBLINGS | 189 | G | 88 | 94.5 | A | 101 | 94.5 | 0.87 (0.65–1.16) | 0.3443 |
| *PLCB2* | rs2229690 | c.1182C>T | MMC | 132 | G | 85 | 66 | A | 47 | 66 | 1.81 (1.27–2.58) | 0.0009* |
| | | | SIBLINGS | 132 | G | 56 | 66 | A | 76 | 66 | 0.74 (0.52–1.04) | 0.0817 |
| *TFAP2A* | rs1675414 | c.46-1620A>G | MMC | 180 | A | 108 | 90 | G | 72 | 90 | 1.50 (1.11–2.02) | 0.0073* |
| | | | SIBLINGS | 184 | A | 93 | 92 | G | 91 | 92 | 1.02 (0.77–1.36) | 0.8828 |

IT, Number of Informative Transmissions; O, Count of Observed Transmissions; E, Count of Expected Transmissions; OR (95% CI), Odds Ratio (calculated as the ratio of transmitted/not transmitted alleles) and 95% Confidence Intervals; $\chi^2$, Chi-Squared Test Value with one degree of freedom; P, P-Value (*<0.05).

**Table 3. Results of the Transmission Disequilibrium Test Analysis in Discordant Sibling Pairs (TDT-DS).**

| GENE | RefSNP | DNA Change | IT | MMC | | SIBLINGS | | OR (CI 95%) | P-value |
|---|---|---|---|---|---|---|---|---|---|
| | | | | T | NT | T | NT | | |
| *PSMB4* | rs4603 | c.701T>C | 165 | 96 | 69 | 66 | 99 | 2.09 (1.31–3.32) | 0.001 |
| *ATIC* | rs2372536 | c.347C>G | 173 | 106 | 67 | 80 | 93 | 1.84 (1.17–2.89) | 0.007 |
| *DKK2* | rs17037102 | c.437G>A | 172 | 101 | 71 | 79 | 93 | 1.67 (1.07–2.63) | 0.023 |
| *MTHFR* | rs1801133 | c.677C>T | 178 | 107 | 71 | 93 | 85 | 1.38 (0.89–2.14) | 0.165 |
| *MTRR* | rs162036 | c.1049A>G | 191 | 110 | 81 | 91 | 100 | 1.49 (0.98–2.28) | 0.065 |
| *SLC7A6OS* | rs3803650 | c.134G>A | 166 | 96 | 70 | 91 | 76 | 1.15 (0.73–1.81) | 0.581 |
| *PSEN2* | rs1046240 | c.261C>T | 178 | 103 | 75 | 82 | 96 | 1.61 (1.04–2.50) | 0.034 |
| *C2CD3* | rs4453265 | c.4923A>G | 189 | 110 | 79 | 88 | 101 | 1.60 (1.04–2.45) | 0.030 |
| *PLCB2* | rs2229690 | c.1182C>T | 132 | 85 | 47 | 56 | 76 | 2.45 (1.45–4.16) | < 0.001 |
| *TFAP2A* | rs1675414 | c.46-1620A>G | 180 | 108 | 72 | 93 | 91 | 1.47 (0.95–2.27) | 0.074 |

IT: Number of Informative Transmissions; T, Number of transmitted alleles; NT, Number of not transmitted alleles, OR (95% CI), Odds Ratio (calculated as the ratio of transmitted/not transmitted alleles), and 95% Confidence Intervals.

folic acid fortification in Mexico at the end of 2001. With these a priori probabilities, we estimated the individual variant and combined risk effect associated with the gene variants showing SSD with the TDT-DS (Table 3). These estimates were based on the transmitted and non-transmitted GRV frequencies to MMC cases and healthy siblings.

The individual and combined conditional probabilities results for the six variants displaying SSD in the TDT-DS analysis are exhibited in the S2 Table. These probabilities concern a zygote carrying the risk variants of these candidate genes. In one variant analysis, the *PLCB2* gene synonymous variant c.1182C>T had the highest risk for MMC with a probability of 1/600 births (1.52 times more than the a *priori* probability of 1/910). However, in the second period (2003–2019), characterized by the folic acid fortification public health program, the same variant's attributable probability of MMC decreased to 1/1172 (1.63 times lower than the initial *a priori* probability). Similarly, the missense variants of *PSMB4*, *ATIC*, and *DKK2* showed increased individual probabilities of 1/626, 1/687, and 1/712 births during the pre-FAFP period. Comparatively, during the post-FAFP, the probabilities of MMC for these gene variants reduced the risk to practically half the pre-FAFP, 1/1223, 1/1343, and 1/1392, respectively (S2 Table).

In the same table, including the six genes, the three missense variants of plus the three synonymous ones of genes, the risk increases to 1/156 births during the pre-FAFP, representing 5.84 times more the a priori risk, being it reduced to 1/304 during the post-FAFP. Call the attention that although variants in genes *MTHFR*, *MTRR*, *SLC7A6OS*, and *TFAP2A* did not show SSD in the TDT-DS, they did show a trend to occur more frequently in MMC cases than in the siblings group. Furthermore, including these four gene variants to the first six, the Bayesian analysis depicted an increased risk for these ten variants in the pre-FAFP of 1/89 births, a 10.2-fold rise compared to the a priori risk of 1/910 births), while during the post-FAFP, it decreased to 1/174, still 5.2-fold higher than the prevalence at birth of 1/910 risk of the pre-FAFP.

A Sankey diagram (Fig 1) can be graphically appreciated, providing a visual perception of the conditional probability of MMC risk increases in the presence of different combinations of the six identified GRVs on the figure's left side. The graph shows that the more risk variants are simultaneously present in a zygote, the greater the risk for an MMC embryo. Some specific gene risk variant combinations present higher hazards, which could be considered the genetic

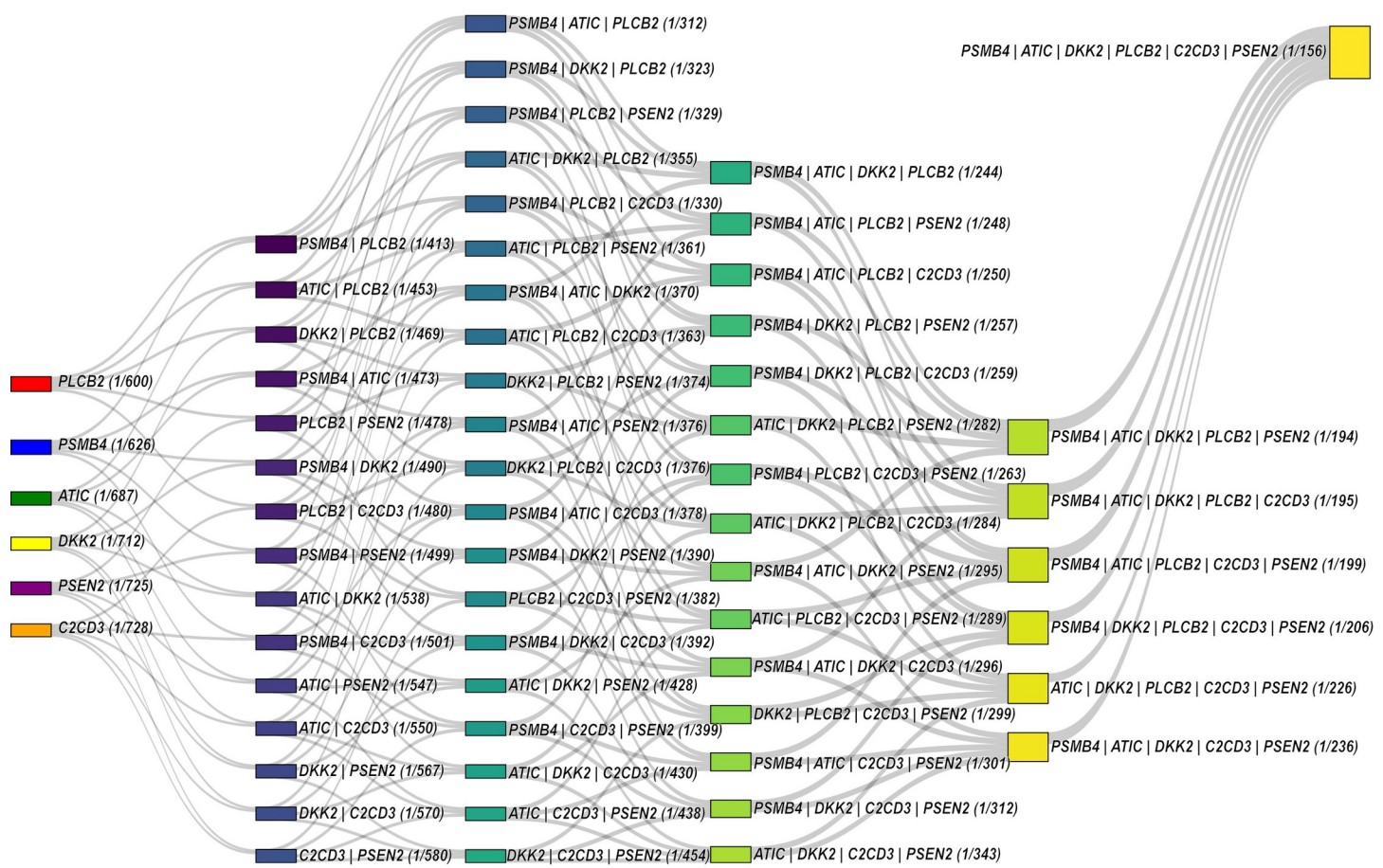

**Fig 1. Sankey diagram.** The figure shows the Bayesian conditional probabilities of gene variant combinations associated with MMC. Each node represents a unique gene or gene combination, colored distinctly to differentiate between individual genes and combinations thereof. The flows between nodes are proportional to the estimated probabilities. This visualization summarizes the interplay between multiple genes and their variants, highlighting the statistically significant combinations identified in the TDT in families with MMC discordant sibling pairs (TDT-DS) analysis.

component of an individualized risk predisposition in families with that candidate gene risk array.

## Discussion

Our results show that the analytical strategy implemented proved valuable for assessing complex genetic disease risk etiology like congenital malformations, allowing this strategy to estimate a series of polygenic risks for MMC in 203 nuclear families with discordant sib-pairs using a Bayesian analytical model [35], to the recognition of different predictable polygenic risk ratios. Given the shared genetic background and the intrauterine environmental similitudes among siblings, discordant pairs facilitate the control of potential confounders, particularly environmental factors, and population stratification.

We identified significant associations of six genes having specific characteristics linked to neural tube closure. *PLCB2*, involved in endocannabinoid signaling, has been linked to NTDs in mice [36]. *PSMB4* participates in protein degradation and *ATIC* in purine biosynthesis. *DKK2* acts in the Wnt signaling pathway regulating embryonic development. *PSEN2* is

essential for β-amyloid production and Notch signaling [37], and *C2CD3* plays a fundamental role in centriole elongation and ciliogenesis [38,39], all functions related to NTD predisposition.

The Bayesian analysis of conditional probabilities, including the group of GRV resulting from TDT-DS analysis, revealed evident MMC occurrence risk changes during two distinct periods. In the initial pre-FAFP (1978–2002), the prevalence of MMC was 1/910 births (1,049 cases in 954,080 live births).[4,5,24] The polygenic combined risk for the six candidate gene variants that showed SSD reached a maximum likelihood of occurrence of 1/156 live births, 5.84 times higher risk for MMC than the RYVEMCE a priori live births in the whole (954,080) population surveyed. Additionally, the risk increases to 1/89, a 10.2-fold increase, if variants of the genes involved in folate metabolism, missense variants c.677C>T of *MTHFR* and 1049G of *MTRR* gene, and synonymous and intronic variants 134A and 46-1620G from *SLC7A6OS* and *TFAP2A* genes respectively are included in the Bayesian analysis. Interestingly, although during the fortification period of 2003–2019, the MMC's prevalence decreased to 0.562/1,000 or 1/1,779 live births (149/265,016 live births), the risk for all these ten variants diminished half to 1/174 live births, yet remaining 5.23-fold increase from the a priori baseline risk of 1/910 live births.

We observed a distinctive behavior in transmitting risk alleles of the missense variant c.701T>C in the *PSMB4* gene encoding a 20S proteasome β subunit within the 203 nuclear families. This variant showed a significant disequilibrium from the expected Mendelian transmission of the risk and wildtype alleles. In particular, it exhibited a significant over-transmission of the risk allele to MMC children but significant under-transmission of the risk allele to healthy siblings, with ORs of 1.39 (1.01–1.92) and 0.67 (0.48–0.92) respectively (Table 2). The essential roles in cell proteolytic degradation ensure the removal of misfolded and damaged proteins that participate in antigenic peptide reactions, suggesting that specific variants in the PSMB4 gene might affect protein degradation during neurogenesis, explaining in part present findings between protein abundance during embryogenesis [40].

The results of the TDT-DS analysis in discordant sib-pairs of variants in genes *PSMB4* (c.701T>C), *ATIC* (c.347C>G), and *DKK2* (c.437G>A) revealed a significant association with MMC, with a significant passing on the risk allele to the affected respect their healthy siblings. Also worthy of note is that no statistically significant higher frequency of risk allele transmissions of *MTHFR*, *MTRR*, and *SLC7A6OS* gene variants was found, suggesting that slight differences collaborate in defining a polygenic risk intensity for MMC depending on their function and population studied [3,41,42]. Furthermore, synonymous variants in *PLCB2*, *PSEN2*, and *C2CD3* genes, which showed SSD, strengthen the importance of considering synonymous variants inclusion in the analysis of association studies, as they may have subtle functional impacts or be in linkage disequilibrium as genome positional markers, offering a further perspective for the interpretation of the estimated polygenic risk assessment.

The Bayesian analysis enables the estimation of risk ratios for various gene variant combinations, assessing the associated risks for the occurrence of a child with MMC, as shown in the S2 Table. The a priori probability established from the RYVEMCE data for isolated MMC from the pre- and post-folic acid fortification periods [4,5] facilitated improved adjustments of the probabilities for polygenic risk estimations. The Sankey diagram's graphical representation describes the cumulative risk observed, highlighting how it increases as more gene risk variants contribute to the polygenic risk estimates. This approach underscores the variable MMC's polygenic predisposition and environmental factors from one case to another, which contribute to a specific risk level profile in certain families. Interestingly, the observed temporal changes in estimated risks emphasize the modifier effect of folic acid diet fortification on the genetic MMC's risk predisposition and the impact of public health preventative health

measures. Continued research is essential to understand better how these genetic effects translate into very early embryo-developmental risk and how they can help in the custom-made prevention of MMC occurrence through appropriate and timely genetic counseling.

In conclusion, polygenic risk assessments through a model based on discordant sibling pairs using the Bayesian method of analysis allow us to identify variable degrees of polygenic risks for MMC in 203 families from 11 different regions of the country, giving an acceptable setting of the country population. In summary, our results using this methodology and Bayesian analysis acknowledge more precise and tailored genetic counseling and *ad hoc* prevention measures based on this genetic predisposition estimates approach than risk assessments founded on empirical data from other populations with much different ethnical and genetic structures than the Mexican population [43,44].

## Supporting information

**S1 Table. Results of McNemar's test and conditional logistic regression in discordant sibling pairs for MMC.**
(XLSX)

**S2 Table. Bayesian polygenic MMC risk estimated considering the TDT-DS conditional probability.**
(XLSX)

**S1 File. Supplementary Bayesian method description.**
(DOCX)

**S2 File. Shiny application code.** Bayes probability calculator.
(R)

**S3 File. Example data.** CSV files for input into the Shiny application.
(CSV)

## Acknowledgments

We sincerely thank the families whose collaboration made this study possible and dedicated physicians and students for their invaluable participation.

## Author Contributions

**Conceptualization:** Adolfo Aguayo-Gómez, Luis Ángel Muñoz-Téllez, Osvaldo M. Mutchinick.

**Data curation:** Adolfo Aguayo-Gómez, Leonora Luna-Muñoz, Yevgeniya Svyryd, Luis Ángel Muñoz-Téllez.

**Formal analysis:** Adolfo Aguayo-Gómez, Yevgeniya Svyryd, Luis Ángel Muñoz-Téllez.

**Funding acquisition:** Osvaldo M. Mutchinick.

**Investigation:** Leonora Luna-Muñoz, Yevgeniya Svyryd, Luis Ángel Muñoz-Téllez, Osvaldo M. Mutchinick.

**Methodology:** Adolfo Aguayo-Gómez, Osvaldo M. Mutchinick.

**Project administration:** Leonora Luna-Muñoz, Yevgeniya Svyryd, Luis Ángel Muñoz-Téllez, Osvaldo M. Mutchinick.

**Supervision:** Osvaldo M. Mutchinick.

**Validation:** Adolfo Aguayo-Gómez, Leonora Luna-Muñoz, Yevgeniya Svyryd, Osvaldo M. Mutchinick.

**Visualization:** Osvaldo M. Mutchinick.

**Writing – original draft:** Adolfo Aguayo-Gómez.

**Writing – review & editing:** Osvaldo M. Mutchinick.

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
