## [Decision Letter · Decision Letter 0]

17 Sep 2024

PONE-D-24-17409Bayesian Polygenic Risk Estimation Approach to Nuclear Families with Discordant Sib-Pairs for MyelomeningocelePLOS ONE

Dear Dr. Mutchinick,

Thank you for submitting your manuscript to PLOS ONE. After careful consideration, we feel that it has merit but does not fully meet PLOS ONE’s publication criteria as it currently stands. Therefore, we invite you to submit a revised version of the manuscript that addresses the points raised during the review process.

We look forward to receiving your revised manuscript.

Kind regards,

Yun Li

Academic Editor

PLOS ONE

Journal Requirements:

Reviewers' comments:

Reviewer's Responses to Questions

**Comments to the Author**

1. Is the manuscript technically sound, and do the data support the conclusions?

Reviewer #1: Partly

Reviewer #2: Yes

Reviewer #3: Yes

2. Has the statistical analysis been performed appropriately and rigorously? 

Reviewer #1: Yes

Reviewer #2: Yes

Reviewer #3: Yes

3. Have the authors made all data underlying the findings in their manuscript fully available?

Reviewer #1: Yes

Reviewer #2: Yes

Reviewer #3: Yes

4. Is the manuscript presented in an intelligible fashion and written in standard English?

Reviewer #1: Yes

Reviewer #2: Yes

Reviewer #3: Yes

5. Review Comments to the Author

Reviewer #1: In this paper, the authors have adopted a Bayesian approach to estimate polygenic risk

and identify the significant genes related to neural tube disorders (NTDs). While the topic

of the paper is significant, the paper should be better constructed and include more details

about the tests, mechanisms used, assumptions, modeling techniques etc. For example:

1. Section 2.3.3: Bayesian Analysis: Bayesian is an estimation approach which allows

one to incorporate prior knowledge about model parameters. It performs estimation

through posterior sampling. However, it still needs a particular model to work on.

The stats section in the paper lacks proper description of the exact model that was

used, the model parameters, what prior were assumed in the Bayesian analysis and

how posterior sampling was done.

2. Line 134: “The analysis were conducted by categorizing gene variants into missense

and non-missense..." - How was this categorization done?

3. “Independent effects are assumed when combining the effects of multiple variants" -

can the authors discuss the practicality of this assumption and what happens when it

is violated.

4. Bayesian Analysis Results: Can the authors provide some measures of model diagnostics and time complexity?

General comment: The authors should include more details and be specific about the

analysis performed for completion

Reviewer #2: The manuscript described a scientifically plausible protocol in qualifying polygenic risk among the target population.

The statistical analysis is performed in accordance with the aim of the study.

Yes.

Yes.

Reviewer #3: The manuscript presents a study that identifies a polygenic risk profile for Myelomeningocele in Mexican families, highlighting significant genetic associations and the impact of folic acid fortification on disease risk. The study employed a family-based analysis using the transmission disequilibrium test and Bayesian methods to examine 656 single nucleotide polymorphisms across 395 genes in 203 Mexican nuclear families with discordant siblings. The findings reveal that 150 genetic risk variants were identified in 10 genes, with TDT analysis highlighting significant statistical differences in allele transmission for ten variants. Further, Bayesian analysis shows a significant reduction in risk probabilities after the folic acid fortification.

This work advances our genetic understanding of MMC and could enhance prevention and screening strategies for MMC and other congenital malformations.

6. PLOS authors have the option to publish the peer review history of their article (what does this mean?). If published, this will include your full peer review and any attached files.

Reviewer #1: No

Reviewer #2: No

Reviewer #3: No

---

## [Author Response · Author response to Decision Letter 0]

23 Oct 2024

October 21, 2024

Rebuttal letter

Dear Editor-in-Chief,

Dear Academic Editor and Reviewers,

We sincerely appreciate all reviewers' insightful feedback, which allowed us to significantly improve the clarity and depth of our manuscript, "Bayesian Polygenic Risk Estimation Approach to Nuclear Families with Discordant Sib-pairs for Myelomeningocele."

In this rebuttal letter, we have carefully addressed all the reviewers' comments and made the corresponding revisions to the manuscript. Specifically, we have clarified the Bayesian framework used, expanded the methodological details, and provided further discussion on the assumptions and computational aspects of the analysis.

We hope the revised manuscript answers the reviewers' comments and suggestions.

Sincerely yours,

Osvaldo M. Mutchinick, MD, PhD

Professor of Medical Genetics

Chief, Department of Genetics

National Institute of Medical Sciences and Nutrition Salvador Zubirán 

Vasco de Quiroga 15, Col. Belisario Domínguez, Tlalpan

14080, Mexico City, Mexico

Phone: (5255) 5487-0900, Ext. 2514/2515

Direct Phone: (5255) 5655-6138

E-mail: osvaldo@unam.mx

Response to Journal Requirements

We have carefully reviewed the journal's additional requirements and have addressed each of them as follows:

 PLOS ONE Style Requirements

We have ensured that the revised manuscript meets PLOS ONE's style requirements.

 Data Availability Statement

We confirm that our submission includes all the raw data required to replicate our study's results. These data are included within the manuscript and its "supporting Information files" to ensure full reproducibility. Should the reviewers or editors require additional data, we agree to provide them upon request.

 Ethics Statement.

We have included a complete ethics statement in the 'Methods' section of the revised manuscript. This statement specifies the name of the Institutional Review Board (IRB) or Ethics Committee that approved the study, the written consent procedures, and any relevant waivers obtained. We hope these revisions meet the journal's requirements, and we remain available for any further clarifications.

Response to reviewers

Reviewer #1: 

Observation 1. Section 2.3.3: Bayesian Analysis: Bayesian is an estimation approach which allows one to incorporate prior knowledge about model parameters. It performs estimation through posterior sampling. However, it still needs a particular model to work on. The stats section in the paper lacks proper description of the exact model that was used, the model parameters, what prior were assumed in the Bayesian analysis and how posterior sampling was done.

Response: 

We have removed the section numbering from the document. The "Bayesian analysis" section, previously identified as 2.3.3, starts on line 126 of the revised manuscript.

We would like to clarify that our approach employs Bayesian principles to estimate the probability of neural tube defects (NTDs) based on allele transmission frequencies. However, it does not involve a formal Bayesian regression model or posterior sampling. Instead, we use a Bayesian probabilistic framework to adjust for known MMC prevalence, which acts as our prior and calculates posterior probabilities of disease given exposure to specific genetic variants. 

Our model framework is based on Bayes' Theorem, where the prior probability is the population prevalence of MMC, and the likelihood is based on the observed allele frequencies in both MMC-affected and non-affected sibs. Specifically, the posterior probability of disease given exposure to a genetic variant is estimated using the following formula:

P("MMC" │"Exposed to TDT transmitted Variant i" )=(f_"MMC-sibs" ^((i) )×P("MMC" ))/(f_"MMC-sibs" ^((i) )×P("MMC" )+f_"Non-affected-sibs" ^((i) )×(1-P("MMC" )) )

Where:

- f_"MMC-sibs" ^((i) ): Is the frequency of the TDT-transmitted allele in MMC-affected sibs.

- f_"Non-affected-sibs" ^((i) ) : Is the frequency of the TDT transmitted allele in Non-affected sibs.

- P("MMC" ) ∶ Is the prevalence of MMC.

Observation 2. Line 134: "The analysis were conducted by categorizing gene variants into missense and non-missense..." - How was this categorization done?

Response: 

We classified variants into two main categories: missense variants and non-missense variants. Missense variants are nucleotide changes that lead to amino acid substitutions, potentially altering protein function. Non-missense variants include synonymous variants, which do not change the amino acid sequence, and intronic variants, which occur in non-coding regions but may affect gene expression regulation. We used the Ensembl Variant Effect Predictor (VEP) for variant annotation. VEP provided comprehensive predictions on the functional impact of each variant, which we used for categorization. When multiple variants were identified within the same gene, we selected the variant with the highest predicted functional impact or most significantly associated with MMC. More detailed descriptions of the categorization process are provided in the supplementary file.

Observation 3. "Independent effects are assumed when combining the effects of multiple variants" - can the authors discuss the practicality of this assumption and what happens when it is violated.

Response: 

The assumption of independent effects simplifies the calculation of polygenic risk by allowing us to combine the effects of multiple variants multiplicatively, as follows:

P("MMC" │"Exposed to variants i,j,…" )=(∏_k▒f_"MMC-sibs" ^((k) ) ×P("MMC" ))/(∏_k▒f_"MMC-sibs" ^((k) ) ×P("MMC" )+∏_k▒f_("Non-affected-sibs" )^((k) ) ×(1-P("MMC" )) )

This assumption holds when the variants are located in different genes or pathways and do not interact biologically. The above is reasonable for variants that are unlinked and functionally independent. However, when variants are in linkage disequilibrium or are part of the same biological pathway, this assumption may be violated, leading to biased estimates. In such cases, interactions between variants could amplify or suppress the individual effects, which our method does not capture. To mitigate this issue, we selected one variant per gene to minimize potential linkage effects and reduce multicollinearity, thus limiting the risk of confounding due to interactions between linked variants. We recognize that this approach may miss some gene-gene interactions and discuss this limitation.

Observation 4. Bayesian Analysis Results: Can the authors provide some measures of model diagnostics and time complexity?

Response: 

Traditional convergence diagnostics are not applicable since we did not employ a formal Bayesian regression model with posterior sampling. Our approach involves direct calculation of posterior probabilities rather than iterative sampling. However, we can discuss the computational complexity of the analysis. The complexity is driven by the combinatorial nature of calculating probabilities for multiple variants. Given (n) variants, the number of combinations we need to evaluate is (2^n-1). For each combination, we compute the posterior probabilities. Time complexity grows exponentially with the number of variants, but we limit the number of variants per analysis for practical purposes to maintain computational feasibility. We also employed progressive computation in our Shiny application to provide real-time feedback on the calculations, improving user experience.

General comment: The authors should include more details and be specific about the analysis performed for completion.

Response: 

We appreciate the feedback and revised the manuscript to include more details about the analysis, especially about the categorization and selection of variants, the probabilistic framework used for Bayesian analysis, and the assumptions of independence. Additionally, we will provide more specific information about the computational aspects of our methodology, including the limitations of the independent effects assumption and the steps we took to mitigate its potential violations. We hope that our responses clarify and address all comments, and we are open to further enriching our manuscript based on any additional suggestions.

Reviewer #2 and Reviewer #3: The authors sincerely appreciate positive comments on the study. We also thank all the reviewers for their time and thoughtful review of our manuscript.

---

## [Decision Letter · Decision Letter 1]

26 Nov 2024

PONE-D-24-17409R1Bayesian Polygenic Risk Estimation Approach to Nuclear Families with Discordant Sib-Pairs for MyelomeningocelePLOS ONE

Dear Dr. Mutchinick,

Thank you for submitting your manuscript to PLOS ONE. After careful consideration, we feel that it has merit but does not fully meet PLOS ONE’s publication criteria as it currently stands. Therefore, we invite you to submit a revised version of the manuscript that addresses a few remaining minor points raised during the review process.

We look forward to receiving your revised manuscript.

Kind regards,

Yun Li

Academic Editor

PLOS ONE

Journal Requirements:

Reviewers' comments:

Reviewer's Responses to Questions

**Comments to the Author**

1. If the authors have adequately addressed your comments raised in a previous round of review and you feel that this manuscript is now acceptable for publication, you may indicate that here to bypass the “Comments to the Author” section, enter your conflict of interest statement in the “Confidential to Editor” section, and submit your "Accept" recommendation.

Reviewer #1: All comments have been addressed

Reviewer #2: All comments have been addressed

Reviewer #3: All comments have been addressed

2. Is the manuscript technically sound, and do the data support the conclusions?

Reviewer #1: Yes

Reviewer #2: Yes

Reviewer #3: (No Response)

3. Has the statistical analysis been performed appropriately and rigorously? 

Reviewer #1: Yes

Reviewer #2: Yes

Reviewer #3: (No Response)

4. Have the authors made all data underlying the findings in their manuscript fully available?

Reviewer #1: Yes

Reviewer #2: Yes

Reviewer #3: (No Response)

5. Is the manuscript presented in an intelligible fashion and written in standard English?

Reviewer #1: Yes

Reviewer #2: Yes

Reviewer #3: (No Response)

6. Review Comments to the Author

Reviewer #1: The authors have answered all questions and made necessary recommended modifications in the paper. Thank you for addressing the comments.

Reviewer #2: This revision addressed most previous comments, and the research was presented in a clear and informative manner. Some figures and tables, however, need to be adjusted in size for better readability (e.g. Line 163-164, the last column of Table 1 showing HGVSp extends out of the page; line 223, figure 1 is not there). Other issues mostly involve small grammar mistakes (e.g. Line 173, a period punctuation mark is missing) and long sentences that are generally acceptable.

Reviewer #3: (No Response)

7. PLOS authors have the option to publish the peer review history of their article (what does this mean?). If published, this will include your full peer review and any attached files.

Reviewer #1: No

Reviewer #2: No

Reviewer #3: No

---

## [Author Response · Author response to Decision Letter 1]

28 Nov 2024

November 28, 2024

Rebuttal letter

Dear Dr. Yun Li, 

Academic Editor

Plos One

We sincerely appreciate the insightful feedback provided by all reviewers and regret the minor issues raised during the review process. We hope that we have fulfilled the journal's requirements through this carefully corrected revision of our manuscript, 'Bayesian Polygenic Risk Estimation Approach to Nuclear Families with Discordant Sib-pairs for Myelomeningocele'.

Sincerely yours,

Osvaldo M. Mutchinick, MD, PhD

Professor of Medical Genetics

Chief, Department of Genetics

National Institute of Medical Sciences and Nutrition Salvador Zubirán 

Vasco de Quiroga 15, Col. Belisario Domínguez, Tlalpan

14080, Mexico City, Mexico

Phone: (5255) 5487-0900, Ext. 2514/2515

Direct Phone: (5255) 5655-6138

E-mail: osvaldo@unam.mx

Response to Journal Requirements

We have carefully reviewed the journal's additional requirements and have addressed each of them as follows:

1. Completeness and correctness of the reference list:

- We realized that references number 4 and 40 were duplicates. Reference 40 (Lines 416-418) was eliminated, and the annotation of the other references was corrected in the text (lines 246, 266, 273, and 299) and the final list of references (lines 419, 421, 425, 429, and 432).

- Reference 5 (lines 315-318) was corrected to meet the journal's guidelines.

- Reference 35 (lines 401-403): the DOI was added, and the hyperlink to the article was eliminated.

2. Revision of the cited articles:

- We reviewed all cited articles by searching in the Retraction Watch Database [1], CrossRef [2], Google Scholar [3], PubMed [4], and directly on the websites of the journals where the articles were published. None of the 44 references appeared retracted.

[1]. Retraction Watch. Retraction Watch Database [Internet]. New York: The Center for Scientific Integrity; c2024 [cited 2024 Nov 26]. Available from: https://retractionwatch.com/the-retraction-watch-database/

[2]. CrossRef. CrossRef Metadata Search [Internet]. Lynnfield: CrossRef; c2024 [cited 2024 Nov 26]. Available from: https://search.crossref.org/

[3] Google Scholar. Google Scholar [Internet]. Mountain View: Google; c2024 [cited 2024 Nov 26]. Available from: https://scholar.google.com/

[4] National Center for Biotechnology Information (NCBI). PubMed [Internet]. Bethesda: U.S. National Library of Medicine; c2024 [cited 2024 Nov 26]. Available from: https://pubmed.ncbi.nlm.nih.gov/

Response to reviewers

Reviewer #2: 

Comment 6. "This revision addressed most previous comments, and the research was presented in a clear and informative manner. Some figures and tables, however, need to be adjusted in size for better readability (e.g. Line 163-164, the last column of Table 1 showing HGVSp extends out of the page; line 223, figure 1 is not there). Other issues mostly involve small grammar mistakes (e.g. Line 173, a period punctuation mark is missing) and long sentences that are generally acceptable."

Response: 

- Regarding Table 1, we retained its original size as it was necessary to ensure clarity and readability, modifying the paper sheet size.

- Regarding Figure 1, the figure was not placed in the main manuscript, as stipulated in the journal's guidelines (https://journals.plos.org/plosone/s/submission-guidelines#loc-references):

 "Do not include figures in the main manuscript file. Each figure must be prepared and submitted as an individual file. … Figure captions are inserted immediately after the first paragraph in which the figure is cited. Figure files are uploaded separately..." 

- To ensure compliance, we checked the figure using the PACE tool (https://pacev2.apexcovantage.com/). The adjustments were as follows: the image file was flattened, dimensions were adjusted to 7.49 in W x 4.69 in H, and the TIF file was converted to a valid TIF format. Following these changes, we uploaded a new figure file.

- We have thoroughly reviewed the manuscript for grammatical errors and addressed several issues, including:

Line 81: The word "off" was deleted

Line 173: Final punctuation corrected.

Line 181: Extra space removed before the number "0.001".

Lines 118, 186, 195, 198, 212, 227, 244, 266, 433: "TDTDS" changed to "TDT-DS" for better clarity. 

S1 File and S2 Table: "TDTDS" changed to "TDT-DS"

Lines 211, 238-241, 250-252, 258, 267, 268, 271, 273, 274: Gene names formatted in italics.

Line 233: Space added after the word "model."

Line 255: "increased" changed to "increase."

Line 281: "pre" changed to "pre-."

Line 296: "ad hoc" formatted in italics.

---

## [Editor Report · Decision Letter 2]

11 Dec 2024

Bayesian Polygenic Risk Estimation Approach to Nuclear Families with Discordant Sib-Pairs for Myelomeningocele

PONE-D-24-17409R2

Dear Dr. Mutchinick,

We’re pleased to inform you that your manuscript has been judged scientifically suitable for publication and will be formally accepted for publication once it meets all outstanding technical requirements.

Kind regards,

Yun Li

Academic Editor

PLOS ONE
---

## [Editor Report · Acceptance letter]

17 Dec 2024

PONE-D-24-17409R2 

PLOS ONE

Dear Dr. Mutchinick, 

I'm pleased to inform you that your manuscript has been deemed suitable for publication in PLOS ONE. Congratulations! Your manuscript is now being handed over to our production team.

Kind regards, 

on behalf of

Dr. Yun Li 

Academic Editor

PLOS ONE